# Multi-Scale Influence Analysis of Urban Shadow and Spatial Form Features on Urban Thermal Environment

**Liqun Lin [1,2], Yangyan Deng [1], Man Peng [3,*], Longxiang Zhen [1] and Shuwei Qin [1]**

[1]  Faculty of Resources and Environmental Science, Hubei University, Wuhan 430062, China; linliqun16@hubu.edu.cn (L.L.); 202121108012722@stu.hubu.edu.cn (Y.D.); 202121108012726@stu.hubu.edu.cn (L.Z.); qinsw@stu.hubu.edu.cn (S.Q.)

[2]  Hubei Key Laboratory of Regional Development and Environmental Response, Wuhan 430062, China

[3]  The State Key Laboratory of Remote Sensing Science, Aerospace Information Research Institute, Chinese Academy of Sciences, Beijing 100101, China

[*]  Correspondence: pengman@aircas.ac.cn; Tel.: +86-15110247276

**Abstract:** In urban thermal environment research (UTE), urban shadows formed by buildings and trees contribute to significant variations in thermal conditions, particularly during the mid-day period. This study investigated the multi-scale effects of indicators, including urban shadows, on UTE, focusing specifically on the mid-day hours. It integrated field temperature measurements and drone aerial data from multiple city blocks. Considering both urban shadows and direct solar radiation, a linear mixed-effects model was employed to study the multi-scale effects of urban morphological indicators. Results showed that: (1) UTE is a multi-scale, multi-factor phenomenon, with thermal effects manifesting at specific scales. Under shadow conditions, smaller scales (10–20 m) of landscape heterogeneity and larger scales (300–400 m) of landscape consistency better explained temperature variations mid-day. Conversely, under direct sunlight, temperature was primarily influenced by larger scales (150–300 m). (2) Trees significantly reduced temperature; 100% tree canopy cover within a 10-m radius reduced air temperatures by approximately 2 °C mid-day. However, there is no significant correlation between temperature and green spaces. (3) Building area and height were significantly correlated with temperature. Specifically, an increase in building area beyond 150 m, especially within a 300-m radius, leads to higher temperatures. Conversely, building height within a 10–20 m range exhibits significant cooling effects. These findings provide crucial reference data for micro-scale UTE investigations during mid-day hours and offer new strategies for urban planning and design.

**Keywords:** building and urban tree; spatial correlation; scale effects; urban shadows; urban planning; urban heat mitigation

## 1. Introduction

Urban heat island (UHI) is a critical research topic in today's urban environments [1,2]. Rising temperatures heighten the risk of heat-related illnesses for urban residents in summer [3,4]. Moreover, increased temperatures escalate air conditioner energy consumption [5] and exacerbate air pollution [6,7], compromising the livability of cities and hindering sustainable urban ecosystem development. Hence, studying the features and patterns of the urban thermal environment (UTE) is crucial for urban planning, environmental protection, heat mitigation, and municipal infrastructure development [8,9].

The UTE is a complex phenomenon characterized by multiple scales and causative factors [10–12]. Research on UTE can be categorized into three levels: macro, meso, and micro, encompassing spatial scales ranging from individual buildings to entire urban areas [13]. As a multifaceted physical phenomenon, UTE is influenced by various factors, including meteorological conditions, solar radiation, and the urban environment [14]. Notably, the land cover plays a crucial role in shaping the form and layout of urban

landscape and acts as a significant determinant of temperature variation [15–17]. Previous studies have shown that the impacts of urban morphology features on UTE vary with scale and the thermal effects of these indicators differ across different spatial scales [18,19]. Specifically, at the micro scale, indicators such as buildings, roads, and trees within the local urban landscape can influence the surrounding temperature [20,21]. Meanwhile, at the meso and macro scales, indicators such as the city's scale, shape, land use type, and natural environment also affect the UHI intensity [22,23]. Hence, temperatures within cities are subject to influences at multiple scales, covering the macro level of the entire city as well as the micro level of individual buildings and vegetation. An in-depth understanding of the interdependencies and correlations across these diverse scales within the UTE is paramount. Such understanding aids in quantifying the scale-dependent impact of factors contributing to the UHI effect, thus offering valuable insights for the formulation of effective mitigation strategies and regulatory measures.

Remote sensing (RS) primarily investigates the UHI phenomenon at the meso and macro scales, which commonly involves analyzing the city and its surrounding areas, with a specific focus on discussing the thermal variations between urban and rural regions [24,25]. However, this approach reveals that characterizing the UHI as an "archipelago" rather than an "island" is more apt [10]. As a result, it falls short of offering a comprehensive understanding of the underlying mechanisms governing the UTE. In recent years, high-resolution RS images have enabled a clearer visualization of land use types, enabling a closer association between spatially continuous land surface temperature and alterations in surface biophysics. This approach better reflects the surface temperature variations resulting from changes in land cover [26–28]. However, it is worth noting that the spatial resolution of thermal infrared images employed for surface temperature estimation remains relatively low (e.g., Landsat 8, 100 m) [29]. At this scale, individual temperature pixels often amalgamate multiple land cover types, thereby limiting the capability of existing RS methodologies to discern the subtle temperature differences caused by landscape heterogeneity at the micro scale [30,31]. In contrast, the concept of Local Climate Zones (LCZs) [32] classifies urban land surfaces into various categories based on differences in surface cover, structure, material composition, and human activities. The temperature differences between different LCZ types are then utilized to estimate urban heat intensity, serving as a foundation for intracity UTE comparisons [33]. Studies have shown that LCZs of the same type consistently exhibit temperature characteristics. However, this classification approach may overlook the microscale landscape heterogeneity intrinsic to a city's three-dimensional structure and surface materials, which significantly influence the accurate assessment of the UTE [34]. Indeed, surfaces of the same type within an urban environment exhibit varying micro scale landscape parameters, thus resulting in temperature variations. For example, temperature fluctuations in parks are influenced by factors such as the size, shape, and type of vegetation patches, while temperature fluctuations in city blocks are determined by building density, height, and layout. However, the distribution of green spaces, building areas, and building heights within a city is not uniform but rather a random combination of natural features and man-made structures at finer scales. The utilization of numerical and physical models (e.g., Envi-met) is a common practice in micro scale UTE research [35]. However, this approach is hindered by its demand for numerous input parameters and its inclination toward simplified modeling techniques, which pose challenges in accurately capturing the true spatial temperature distribution. Consequently, this limitation obstructs the comparative analysis of intra-urban thermal environments. Therefore, current research on UTE spanning from the micro to meso and macro scales falls short of achieving a comprehensive integration of these diverse scales. Analyzing UTE exclusively at a single scale neglects a thorough assessment of scale effects and spatial correlations among influencing factors, thus resulting in a dearth of quantitative insights elucidating how multiscale spatial indicators collectively impact temperatures within urban areas.

Furthermore, despite extensive research on the influence of urban landscape heterogeneity on local temperature variations at specific spatial scales, there remains a scarcity of quantitative investigations specifically addressing the significant effects of shadows cast by three-dimensional objects, such as buildings and trees, on UTE and its associated influencing factors [36]. Buildings are crucial components of the urban structure, wielding a substantial influence on UTE by shaping the reflection of solar radiation and the dispersion of heat within urban areas [37]. Several studies have examined the impact of buildings on urban temperature, including the relationship between 2D/3D urban spatial form indicators, such as building area and building height, and UTE [16,38–40]. While buildings constitute the principal drivers of UTE, it is noteworthy that building shading can ameliorate the impact of direct sunlight, consequently diminishing the effect of thermal radiation on the surrounding environment and leading to temperature reduction. However, in practical studies, there is a limited consideration of the heating effect of buildings and the cooling effect of shadows together, which may impede the comprehension of the potential influences of other variables on temperature.

Urban trees, in addition to buildings, assume a pivotal role in temperature mitigation through various mechanisms such as intercepting incoming solar radiation, providing direct surface shading, and reducing ambient temperature through evapotranspiration [41,42]. However, it is noteworthy that in many studies investigating the UTE, trees are often grouped together with other forms of vegetation, such as grasslands and shrubs [10]. Although trees have a more pronounced cooling effect compared to low-lying vegetation, the understanding of the extent and scale of tree effects on urban micro scale temperature is hindered by the lack of extensive, fine-grained temperature data as well as the absence of real-time ground coverage data of tree shadows. Therefore, in the process of quantifying the influence of urban spatial indicators on temperature, it is imperative to fully account for the effects of shadows cast by both buildings and trees.

In summary, to comprehend the patterns of urban temperature distribution, it is imperative to take into account a multitude of influencing factors and employ multiscale analysis methods to unravel the underlying spatial correlations. Only by investigating the multiscale framework of UTE based on the heterogeneity of the urban landscape can a more comprehensive and profound understanding of the UTE generation mechanism and temperature distribution variation patterns be achieved.

Our research endeavors are geared towards exploring phenomena within the urban thermal environment across multiple scales, with a keen focus on discerning how different spatial scales influence temperature distribution. We investigate micro scale factors, such as individual buildings and trees, as well as meso/macro scale factors, including street-level regions. Combining remote sensing data and ground-level temperature observations, we employ linear mixed-effects models to investigate how urban spatial indicators, building characteristics, and tree shading affect temperature distribution within the city. We analyze the independent effects of different scales on temperature and reveal their integrated impact. Moreover, we quantify the range and intensity of influence of urban spatial indicators, building characteristics, and tree shading on temperature variations. These comprehensive insights enrich our understanding of urban thermal complexity and offer valuable strategies for future urban planning and design.

## 2. Study Area and Data

### 2.1. Study Area

Wuhan, located in central China, serves as a representative of major cities in the region (Figure 1a), with a population of 13,648,900, of which 11,541,500 reside in urban areas. Wuhan has a subtropical monsoon climate, characterized as humid, with the highest recorded extreme high temperature reaching 41.3 °C on 10 August 1934 [43]. The period from June to September registers the highest temperatures of the year. Over several decades of urbanization, an obvious UHI phenomenon has emerged. The temperature measurements for this study primarily span four functional zones within the city: university

campus ($Z_1$), residential area ($Z_2$), park ($Z_3$), and commercial center ($Z_4$) (Figure 1b). The land cover types mainly consist of buildings, green space (comprising grass and trees), and impervious surfaces, but the proportions of these land cover types vary across the four zones (Figure 1c). The key characteristics of each zone are as follows: $Z_1$ features relatively few buildings but boasts an abundance of trees, $Z_2$ exhibits a high density of both trees and buildings, $Z_3$ encompasses abundant trees but fewer buildings, and $Z_4$ has fewer trees and a high density of buildings. Among these zones, $Z_4$ carries the highest proportion of buildings, $Z_3$ has the highest proportion of trees and green spaces, and $Z_1$ has the highest proportion of impervious surfaces.

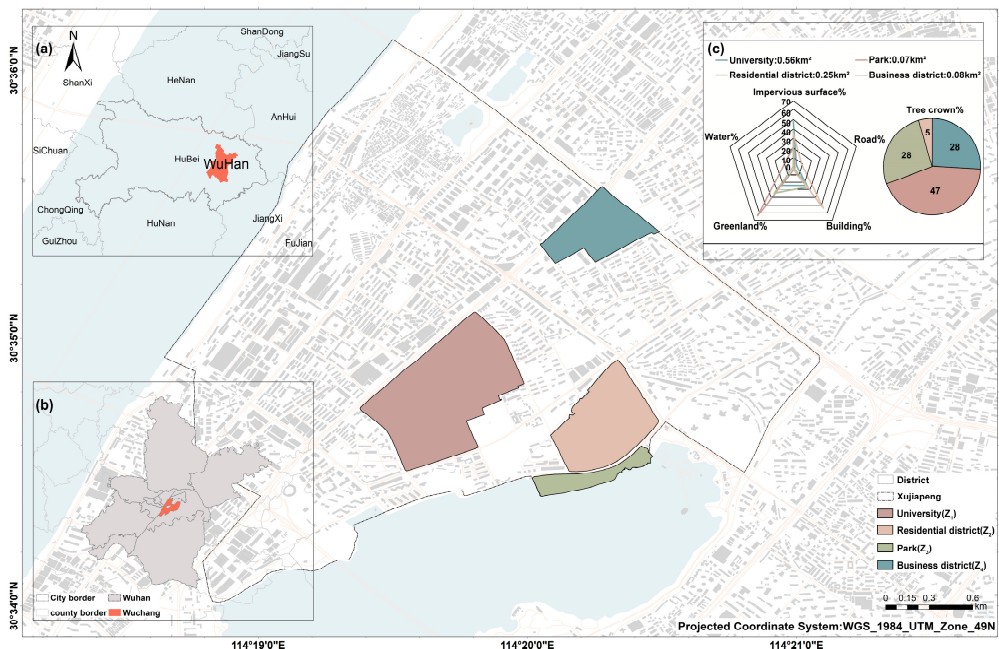

**Figure 1.** Study area. (**a**) The geographical location of Wuhan. (**b**) The geographical location of the four functional zones. (**c**) The size of the area of the four functional zones and the proportion of their land cover types and canopy widths.

### 2.2. Collection of Temperature Measurements and UAV Data Acquisition

During the three days of 29 August, 30 August, and 17 September 2021, temperature and location data were simultaneously collected from different points within the four areas. These measurements were conducted from 12:00 to 13:30 using temperature recorders (Omega H3009) and global positional system (GPS) devices. The temperature measurements were annotated to indicate whether they were taken in direct sunlight or shaded areas. The thermometer was positioned at a height of 0.8 m above the ground to ensure minimal interference and a stable measurement environment. Temperature readings were obtained after allowing the instrument to stabilize for one minute under calm wind conditions.

During the three sampling sessions, a total of 448 temperature records were obtained from 182 sampling points (as detailed in Table 1). Specifically, under sunny conditions, 45 points were sampled on the first day, resulting in 45 records; 47 points were sampled on the second day, resulting in 47 records; and 30 points were sampled on the third day, resulting in 30 records. Among them, 25 points remained in the same location throughout the three days, accumulating a total of 75 records. The remaining points were randomly observed on one or two days, resulting in 47 records. Under shade conditions, a total of 326 records were collected, with 106, 111, and 109 points sampled, respectively. Among them, 94 points were measured on all three days, contributing to a total of 282 records, while the remaining points accounted for 44 records.

**Table 1.** Summary of three sampling sessions.

| Sampling Date | Records Different Location | Sunlight Records from Same Location | Total Records | Records Different Location | Shadow Records from Same Location | Total Records |
|---|---|---|---|---|---|---|
| 29 August 2021 | 20 | | 45 | 12 | | 106 |
| 30 August 2021 | 22 | 25 | 47 | 17 | 94 | 111 |
| 17 September 2021 | 5 | | 30 | 15 | | 109 |
| total | | | 122 | | | 326 |

In September and November 2021, a DJI-M300 RTK UAV equipped with a ZENMUSE P1 lens (f = 35 mm) was deployed to capture vertical images of the experimental area. The flight parameters included an 80% heading overlap and a 70% side overlap. The absolute flight altitude was 500 m, and the collection ground range was 7 km$^2$.

## 3. Method

The framework of multi-scale influence analysis is illustrated in Figure 2. First, digital orthophoto model (DOM) and high-precision digital surface model (DSM) are generated from UAV images. Second, based on DSM, DOM, and temperatures data, 17 circular buffers surrounding each temperature measurement point are determined, the land cover data are visually interpreted based on DOM, and eight indicator parameters can be obtained. Third, a linear mixed-effects model (LME) analysis is conducted on the data, and the quality is assessed. Finally, scale analysis is performed based on three distinct experiment results.

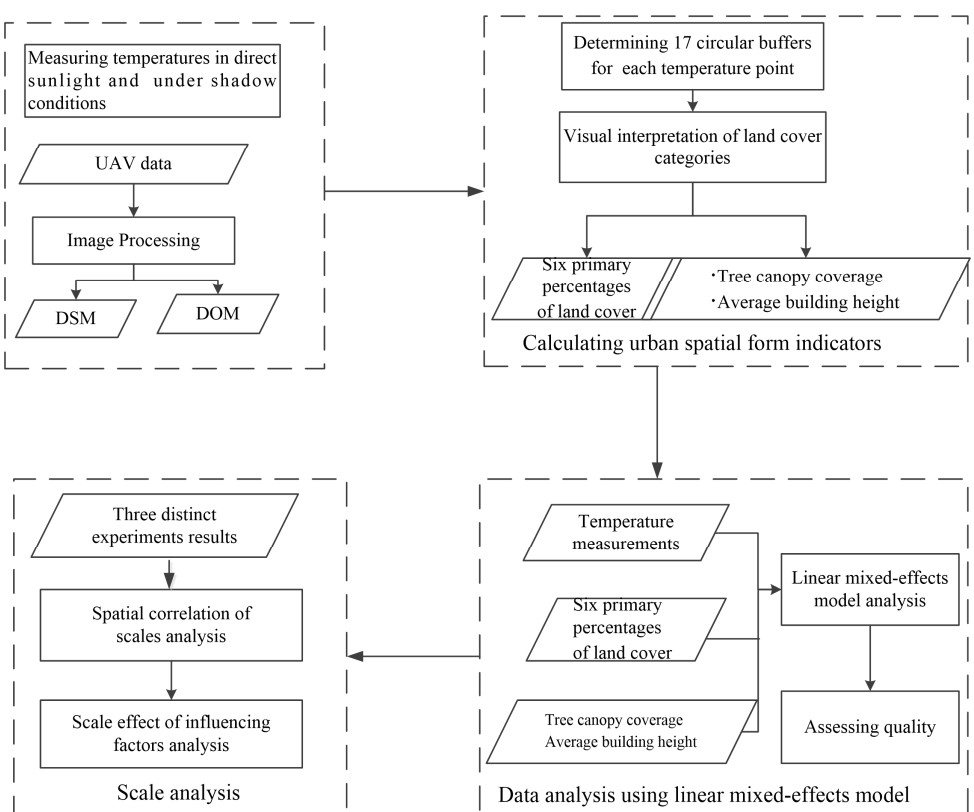

**Figure 2.** Framework of multi-scale influence analysis.

### 3.1. UAV Image Processing

DJI Map software was utilized to process the UAV images, resulting in the generation of a high-resolution DOM and high-precision DSM of the study area. The ground resolutions of the UAV images, without ground control points, were 4.7 cm for the DOM, 9.4 cm for the DSM, and 0.5 m for elevation accuracy.

### 3.2. Spatial Form Indicators

We calculated urban spatial form indicators within 17 circular buffers surrounding each temperature measurement point with radii ranging from 2.5 m to 500 m. These buffers are denoted as D2.5, D5, D10, D20, D30, D40, D50, D60, D70, D80, D90, D100, D150, D200, D300, D400, and D500. Within each of these circular buffers, we derived eight indicator parameters, treating each buffer as a distinct scale. These parameters included six primary percentages of land cover, specifically: buildings (BldgL), green spaces (GreenL), impervious surfaces (ImpS), roads (RoadD), water bodies (Water), and bare ground (BareL). Additionally, we calculated the percentage of tree canopy coverage (TreeC) and the average building height (BldgH).

The land cover data were visually interpreted based on DOM data to ensure the utmost accuracy in data analysis. It was classified into six categories: impervious surfaces (comprising open squares, parking lots, sidewalks, and other cement surfaces), green spaces (with a specific emphasis on grass and shrubs), buildings, roads (encompassing municipal roads, excluding small roads within the area), bare land, and water bodies.

Tree canopy coverage was extracted using absolute height values from the DSM within the 4–28 m range. It was further refined using DOM data to eliminate any misclassifications of building data within this height range. Building height was derived based on the land cover classification data and DSM data, using Geographic Information System spatial analysis to extract the corresponding values.

### 3.3. Data Analysis

In the experiment, ANOVA testing was used to analyze the temperature data collected from the samples. This analysis was primarily conducted to investigate whether there were significant differences in the average temperature values among different regions. Subsequently, the Least Significant Difference (LSD) test was applied for post hoc comparisons between regions to determine if there were notable statistical variations in the average temperature values among them. Furthermore, a LME analysis was conducted on the data, following the approach described by [44,45]. This analysis considered the repeated measurements of data and the spatial correlation between sampling points. By employing this method, we could avoid the issue of losing important information that may occur when using simple mean-based analyses, thereby resulting in more robust and dependable conclusions.

The LME model serves as an extension of the linear model, where the traditional general linear model assumes fixed effects for the independent variables. Independence of samples is one of the key assumptions in linear models, which requires that each data point originates from a distinct population. However, due to the presence of repeated measurements and spatial correlation among the sampling points, the data may not adhere to the independence assumption. The LME model addresses this issue by incorporating random effects, allowing for the inclusion of the non-independence properties of the samples and improving the model fit.

Formulated with each sampling point as the primary unit of analysis, the LME model is expressed as follows:

$$Y_{ti} = \beta_1 * X_{ti}^{(1)} + \beta_2 * X_{ti}^{(2)} + \cdots + \beta_p * X_{ti}^{(p)} \quad (fixed)$$
$$+ u_{1i} * Z_{ti}^{(1)} + \cdots + u_{qi} * Z_{ti}^{(q)} + \varepsilon_{ti} \quad (random)$$

where $Y_{ti}$ represents the temperature measurement for the *t*-th subject (sample) on the *i*-th day; During the fixed part, $X_{ti}^{(P)}$ represents the *p*-th urban spatial form indicator on the *i*-th day for the *t*-th subject. $\beta_p$ is the corresponding coefficient; $\varepsilon_{ti}$ represents the random error in the temperature of each sample point. Additionally, *U* and *Z* represent the random variation in each independent variable. However, only the fixed effects and random intercepts are considered in the experimental analysis, and *U* and *Z* of the random coefficient slope are not considered.

The experimental data were modeled using the R language [45]. The lme4 and lmerTest toolkits were utilized for stepwise regression of the LME model, while the Car package calculated the multicollinearity (VIF). The LME model's quality was assessed using the Akaike information criterion (AIC), where smaller values indicate higher quality. Furthermore, it is worth noting that the data analysis process could also be performed in SPSS software.

### 3.4. Scale Analysis

We divided all the sampling points into two categories: data collected in direct sunlight and data gathered under shadow conditions. For each set of data, we conducted three distinct experiments:

Experiment 1 (Exp. 1: Single scale): Only the spatial form indicators of a single scale is used to analyze the temperature change characteristics.

Experiment 2 (Exp. 2: Potential Multiscale): On the basis of Exp. 1, Exp. 2 incorporates the zone attribute ($Z_1$, $Z_2$, $Z_3$, and $Z_4$) as a control variable. This approach allows us to investigate the explanatory capability of a single-scale spatial form indicator regarding temperature variations within the constraints of a larger spatial context.

Experiment 3 (Exp. 3: Multiscale effect): Exp. 3 aims to identify scales that can characterize the zones defined in Exp. 2 (which are of man-made definition and have varying size ranges). Within the set of 17 existing scales, we select any two scales and employ their spatial form indicators to model the temperature data. This analysis explores the explanatory power of different combinations of two scales' spatial form indicators on temperature variations, aiming to understand the scale–space relationship of UTE. Considering all possible pairs of scales, Exp. 3 will generate a total of $C_{17}^2 = 136$ models fitting results.

Due to the temperature difference over the three-day sampling period, the sampling date (time = 0829, 0830, and 0917) was included as a control variable in each experimental group's model calculation. The variables for each experimental group are listed in Table 2. To ensure the quality of the mixed model, VIF between variables needed to be less than 5, on the basis of a correlation analysis. A stepwise regression approach was employed in the experiment, resulting in some indicators not being included in the model calculation. Therefore, the actual number of indicators included in the model calculation was lower than the total number of variables listed in Table 2.

**Table 2.** Experiment setup.

| | Attribute | | | |
| | Urban Spatial Form Indicators | Control Variables Time | Zone | Total Number of Variables |
|---|---|---|---|---|
| Exp. 1 | 8 | √ | -- | 9 |
| Exp. 2 | 8 | √ | √ | 10 |
| Exp. 3 | 16 | √ | -- | 17 |

## 4. Results

### 4.1. Data Processing Results

The distribution maps of DOM and DSM obtained from UAV data processing are shown in Figure 3a,b. Land cover data, tree canopy distribution data, and building height data are presented in Figure 3c–e, respectively. Additionally, four sampling points with different radius ranges are selected within $Z_1$ and $Z_4$, and the corresponding images are displayed in Figure 4.

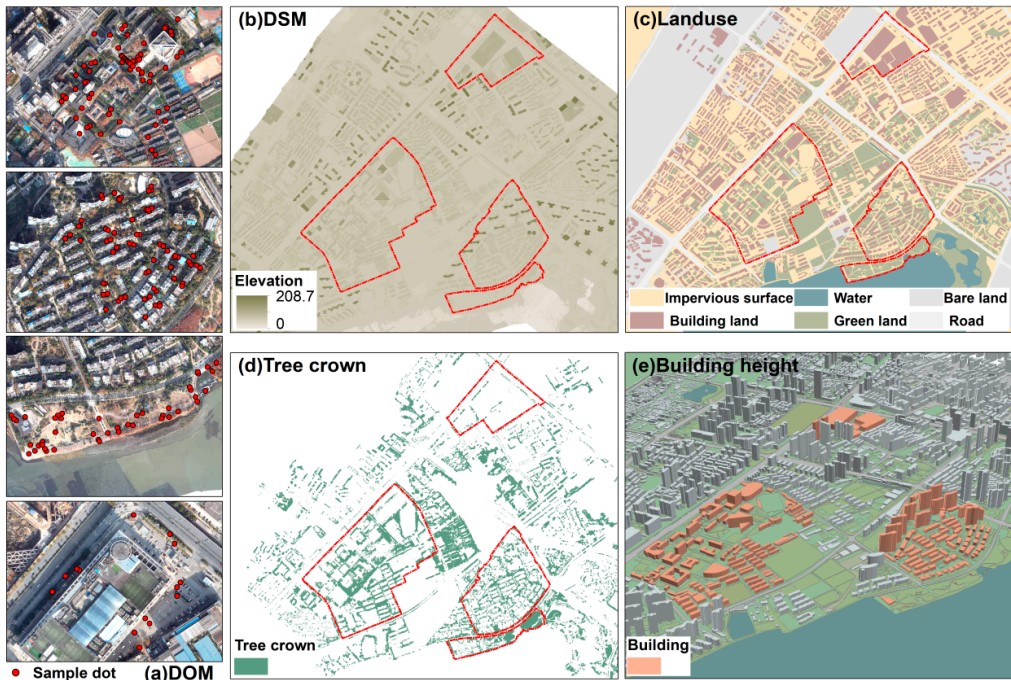

**Figure 3.** Data processing results. (**a**) The DOM obtained by UAV data processing has a ground resolution of 4.7 cm and distribution of the locations of manual field sampling points in the four functional zones. (**b**) DSM obtained from UAV data processing, with a ground resolution of 9.4 cm. (**c**) Land cover data. (**d**) Tree canopy distribution data. (**e**) Building height data.

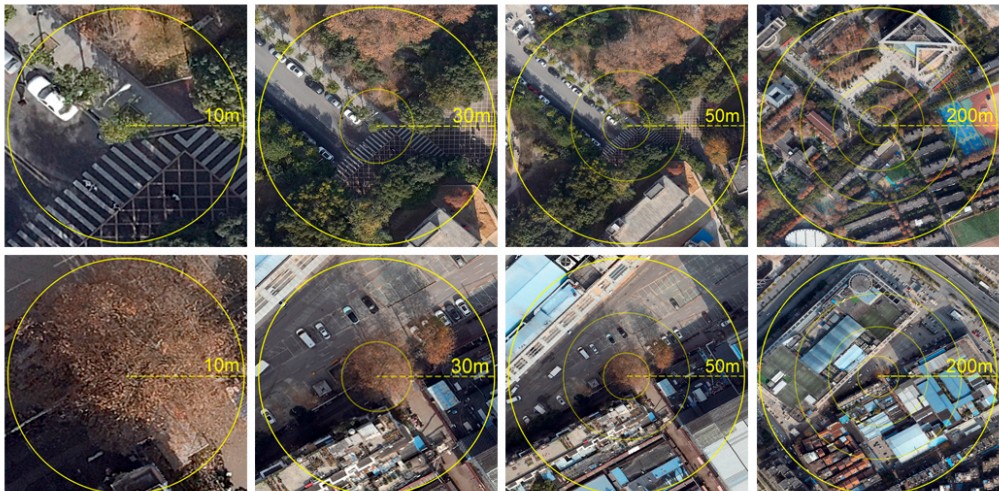

**Figure 4.** Buffer radius schematic. Select one sample point each in $Z_1$ and $Z_4$ to list the DOM covered by the four scales of D10, D30, D50, and D200 (e.g., D10 represents a circular area centered around the sample point with a radius of 10 m).

### 4.2. Descriptive Statistics

Significant temperature variations were observed within the urban area (Figure 5a–d). ANOVA statistical analysis revealed significant differences in temperatures between the sun and shade conditions in all three sampling periods ($p < 0.01$, Figure 5f). Additionally, there were significant differences in temperatures among the four functional zones ($Z_1$–$Z_4$) under both sunlight and shade conditions ($p < 0.01$) (Figure 5e). Under shade conditions, the average temperatures showed the following trend: $Z_4 > Z_2$ and $Z_3 > Z_1$. However, under sunlight, the average temperatures showed the following trend: $Z_4 > Z_2 > Z_3$ and $Z_1$. This indicates the presence of temperature variations among different zones.

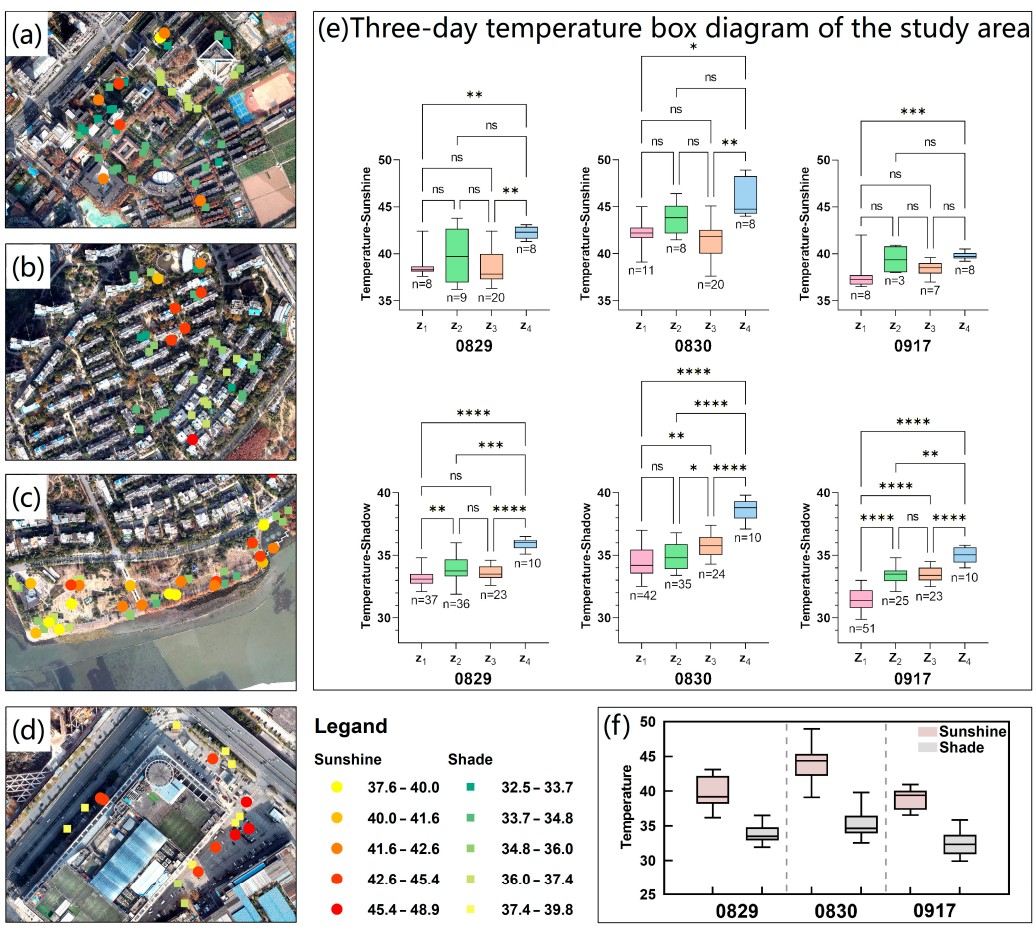

**Figure 5.** Statistical chart of measured temperature distribution. (**a**–**d**) represent the temperature level distribution within the university campus ($Z_1$), residential area ($Z_2$), park ($Z_3$), and commercial center ($Z_4$) under both direct sunlight and shaded conditions, respectively. (**e**) displays the LSD comparison of temperatures among different functional zones at different time points, where * indicates statistical significance with $p < 0.05$ (* The more you have, the more significant it is.), ns indicates non-significance with $p > 0.05$, and "n" represents the number of sampling points. (**f**) presents temperature box plots under both sunlight and shaded conditions (ANOVA, $p < 0.01$).

Table 3 describes the temperatures recorded at Time = 0830. Under shade conditions, the mean and standard deviation (SE) for each zone ($Z_1$–$Z_4$) were $34.5 \pm 1.24$, $34.9 \pm 1.07$, $35.8 \pm 0.85$, and $38.6 \pm 0.91$, respectively. However, under sunlight, variations in ground-level solar irradiance due to cloud cover resulted in larger SE values. These observations indicate that within the same zone, there is a similarity in temperature patterns, while temperature variations exist within the zone.

**Table 3.** Description of the recorded temperature for Time2 = 0830.

| Zone | Sunlight | | | | Shadow | | | |
|------|------|------|---------|---------|------|------|---------|---------|
| | Mean | SE | Minimum | Maximum | Mean | SE | Minimum | Maximum |
| $Z_1$ | 42.2 | 1.76 | 39.1 | 45 | 34.5 | 1.24 | 32.5 | 37 |
| $Z_2$ | 43.7 | 1.66 | 41.5 | 46.4 | 34.9 | 1.07 | 33.4 | 36.8 |
| $Z_3$ | 41.5 | 1.83 | 37.6 | 45.1 | 35.8 | 0.85 | 34.3 | 37.4 |
| $Z_4$ | 45.9 | 2.05 | 44.1 | 48.9 | 38.6 | 0.91 | 37.1 | 39.8 |

However, the subjective zoning of $Z_1$–$Z_4$ functional areas with inconsistent sizes is inadequate for representing urban temperature characteristics without fixed range sizes. To account for complex land use and landscape heterogeneity, a universally applicable

scale is needed to reflect temperature similarity in large-scale urban areas and variability in small-scale local areas.

### 4.3. Spatial Correlation of Scales

Under shadow condition, the D40–D50 and D300–D500 scales show higher quality (Figure 6a), while under sunlight condition, D150–D300 performs better (Figure 6b) in Exp. 1, indicating their effectiveness in explaining urban temperature variations.

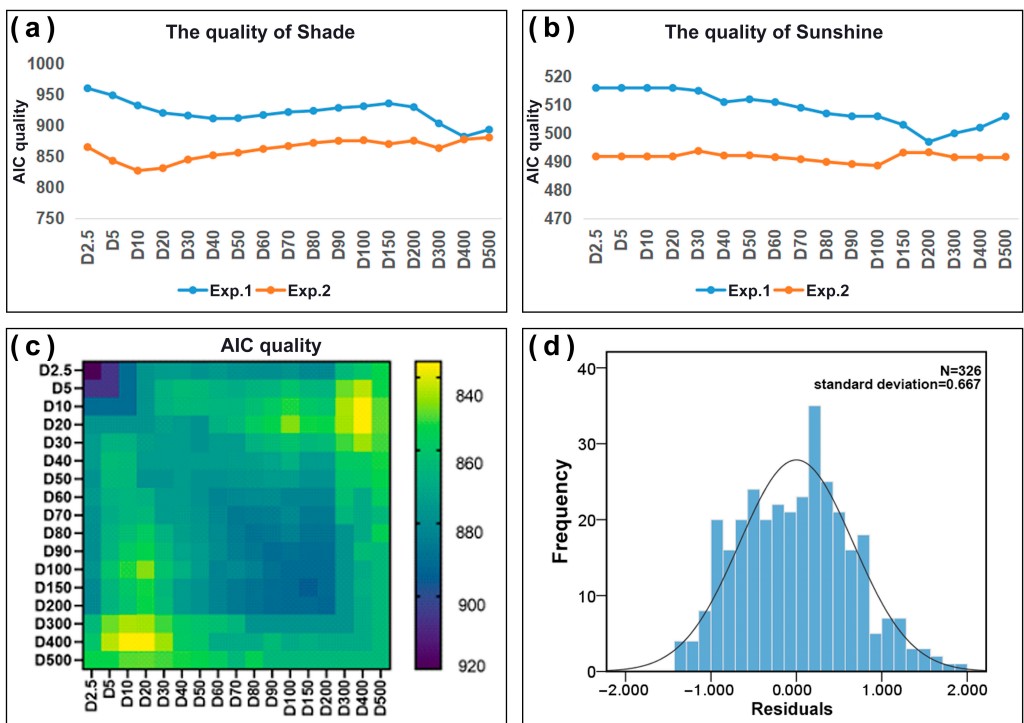

**Figure 6.** Model quality distribution. (**a**) Under shadow conditions, Exp. 1 and Exp. 2 regress the AIC distribution of model quality values. (**b**) Under direct solar light, Exp. 1 and Exp. 2 regress the AIC distribution of model quality values. (**c**) Exp. 3 displays the AIC distribution of model quality values for multiscale regression under shadow conditions. (**d**) Under shadow conditions, Exp. 3 demonstrates the distribution of residuals for the multiscale D10 + D400 regression model.

In Exp. 2, the inclusion of the zone attribute as a control variable, which represents the influence of larger-scale factors, revealed that smaller scales D10 and D20 exhibit relatively higher quality (Figure 6a). Furthermore, the AIC quality measure improved from 912 for D40 in Exp. 1 to 827 for D10 in Exp. 2. However, under sunlight conditions, the quality distribution curve across the entire scale range shows minimal fluctuations (Figure 6b).

The experimental results above demonstrate that incorporating the zone attribute, representing potential large-scale influence, can improve quality. This multiscale effect is particularly pronounced under shadow conditions. However, under sunlight conditions, where continuous heating from the sun is present, no significant variation patterns are observed. Therefore, with a focus on shadow conditions and in order to find a quantitative measure for describing the large-scale range, a multiscale comprehensive experiment (Exp. 3) was conducted.

In Exp. 3, we combined different spatial form indicators from any two scales and discovered some relationships among the scales through a heatmap of fitting quality for 136 mixed models (Figure 6c). Compared with the results of Exp. 1, the results of Exp. 3 demonstrate that certain combinations of scales significantly improved the model fit. Among them, the combination of D10 and D400 exhibited the highest quality at 843, approaching the maximum quality of 829 in Exp. 2. Compared to the single-scale analysis in Exp. 1, where D40 (AIC of 912) and D400 (AIC of 882) were considered,

there was a substantial improvement in quality. Specifically, under shadow conditions, the combinations of small scales D10 and D20 with different large scales (D200, D300, D400, and D500) had relatively higher fitting quality. The fitting quality for D10 was 888, 855, 843, and 863, while D20 had slightly lower quality at 875, 856, 845, and 863, respectively. Figure 6d presents a histogram of residuals predicted by the multiscale analysis model, indicating that the majority of residual values are close to zero, with a peak around zero (at 0.28). Additionally, we superimposed a normal curve on the histogram, and the results indicate that the residual distribution closely approximates a normal distribution. This suggests that the model performs well in fitting the data, with residuals uniformly distributed around zero. Thus, it indicates that the analysis of multiple scales can better fit the temperature distribution than a single scale under shadow conditions. Therefore, it can be concluded that the large scales of D300–D400 m under shadow conditions in Exp. 3 can achieve an influence similar to that of the zone variables in Exp. 2. This suggests that the combined effects of small scales (D10–D20 m) and large scales (D300–D400 m) in spatial form indicators can better explain the temperature variation within the urban environment.

### 4.4. Scale Effect of Influencing Factors

(1)  Under sunlight conditions

Based on the results (Table 4), higher precision is observed within the D150–D400 scale under sunlight conditions. As for individual variables, only road distance shows a significant impact on temperature at D2.5–D20, while impervious surfaces and building area positively correlate at D30. Tree canopy and water bodies consistently lead to temperature reduction at D40–D100. Beyond D150, building area has a significant aggregating effect on temperature rise ($p < 0.01$).

**Table 4.** Single scale analysis under sunlight conditions: Stepwise linear regression results with significant parameter coefficients and AIC quality for each model.

| Scale | INT * | Time1 | Time2 | RoadD | TreeC | Water | BldgL | ImpS | AIC | Input Variables |
|-------|-------|-------|-------|-------|-------|-------|-------|------|-----|-----------------|
| D2.5  | 39.45 | 1.09  | 4.52  | −0.003 |      |       |       |      | 516 |                 |
| D5    | 39.45 | 1.09  | 4.52  | −0.003 |      |       |       |      | 516 | RoadD           |
| D10   | 39.45 | 1.09  | 4.52  | −0.003 |      |       |       |      | 516 |                 |
| D20   | 39.45 | 1.09  | 4.52  | −0.003 |      |       |       |      | 516 |                 |
|       |       |       |       |       |       |       |       |      |     | RoadD           |
|       |       |       |       |       |       |       |       |      |     | Water           |
| D30   | 36.72 | 1.16  | 4.58  |       |       |       | 0.034 | 0.030 | 515 | BldgL          |
|       |       |       |       |       |       |       |       |      |     | GreenL          |
|       |       |       |       |       |       |       |       |      |     | ImpS            |
| D40   | 39.87 | 1.13  | 4.55  |       | −0.044 | −0.046 |      |      | 511 | Road            |
| D50   | 39.99 | 1.16  | 4.57  |       | −0.048 | −0.045 |      |      | 509 | TreeC           |
| D60   | 39.99 | 1.18  | 4.59  |       | −0.047 | −0.045 |      |      | 509 | Water           |
| D70   | 40.09 | 1.21  | 4.61  |       | −0.051 | −0.045 |      |      | 507 | BldgL           |
| D80   | 40.17 | 1.23  | 4.62  |       | −0.053 | −0.046 |      |      | 506 | BldgH           |
| D90   | 40.23 | 1.23  | 4.63  |       | −0.055 | −0.047 |      |      | 505 | ImpS            |
| D100  | 40.23 | 1.23  | 4.63  |       | −0.055 | −0.047 |      |      | 504 | GreenL          |
| D150  | 37.02 | 1.12  | 4.53  |       |       |       | 0.065 |      | 501 |                 |
| D200  | 36.72 | 1.19  | 4.53  |       |       |       | 0.077 |      | 497 | Road            |
| D300  | 36.61 | 1.24  | 4.55  |       |       |       | 0.078 |      | 500 | TreeC           |
| D400  | 36.42 | 1.28  | 4.58  |       |       |       | 0.086 |      | 502 | BldgL           |
| D500  | 36.31 | 1.32  | 4.60  |       |       |       | 0.093 |      | 506 | ImpS            |

* The INT (Intercept) represents the model's predicted value when all independent variables are zero. It serves as the baseline prediction. In each model, Time3 = 0907 was chosen as the reference category, while Time1 = 0829 and Time2 = 0830 were compared to Time3. The numerical values for Time1 and Time2 in the model represent the increments relative to Time3 when all other conditions are held constant.

Specifically, within the D40–D100 scale, a 10% increase in tree canopy coverage leads to a temperature reduction of approximately 0.44–0.55 °C, and water bodies cause a temperature decrease of around 0.45 °C. At the D30 scale, a 10% increase in building area and

impervious surfaces raises the temperature by 0.3 °C. However, in the D150–D500 scale, a 10% increase in building area results in a temperature rise ranging from 0.65 °C to 0.93 °C, indicating a warming effect beyond the critical range of 150 m. Analyzing the temperature distribution in different zones (Figure 3), $Z_1$ with dense tree coverage exhibits an average temperature of 42.2 °C (ranging from 39 °C to 44.1 °C), while $Z_4$, dominated by buildings, has an average temperature of 45.9 °C (ranging from 44 °C to 48.9 °C). These temperature variations under solar conditions are primarily influenced by building area beyond 150 m.

As the multi-scale effect under the solar condition is not significant, it was not further discussed.

(2)    Under shadow condition

In Exp. 1 (single-scale), different significant variables were observed at different scales under shadow condition. The tree canopy exhibited a significant cooling effect within the range of D2.5 to D200. The building area showed a significant heating effect beyond the critical range of D150. Building height significantly reduced the temperature at the small scale (D20–30), while it significantly increased temperature at the large scale (D400–D500). Impervious surfaces significantly decreased temperature in the range of D300–D500 ($p < 0.01$, Table 5).

**Table 5.** Single scale analysis under shadow conditions: Stepwise linear regression results with significant parameter coefficients and AIC quality for each model.

| Scale | INT. | Time1 | Time2 | TreeC | BldgH | BldgL | ImpS | AIC | Input Variables |
|---|---|---|---|---|---|---|---|---|---|
| D2.5 | 33.52 | 1.18 | 2.67 | −0.013 | | | | 960 | TreeC |
| D5 | 33.57 | 1.19 | 2.67 | −0.017 | | | | 949 | TreeC GreenL |
| D10 | 33.73 | 1.19 | 2.68 | −0.024 | | | | 933 | ImpS |
| D20 | 34.06 | 1.19 | 2.67 | −0.032 | −0.014 | | | 921 | TreeC GreenL |
| D30 | 34.23 | 1.18 | 2.65 | −0.040 | −0.012 | | | 916 | ImpS BldgH |
| D40 | 34.32 | 1.16 | 2.64 | −0.051 | | | | 912 | TreeC |
| D50 | 34.44 | 1.16 | 2.64 | −0.056 | | | | 912 | GreenL |
| D60 | 34.51 | 1.17 | 2.65 | −0.060 | | | | 917 | ImpS |
| D70 | 34.55 | 1.17 | 2.66 | −0.062 | | | | 922 | |
| D80 | 34.61 | 1.17 | 2.66 | −0.065 | | | | 924 | |
| D90 | 34.57 | 1.17 | 2.65 | −0.064 | | | | 929 | TreeC |
| D100 | 34.53 | 1.16 | 2.65 | −0.064 | | | | 931 | GreenL |
| D150 | 33.96 | 1.15 | 2.64 | −0.057 | | 0.014 | | 936 | BldgL |
| D200 | 33.84 | 1.15 | 2.64 | −0.060 | | 0.023 | | 930 | |
| D300 | 33.43 | 1.11 | 2.61 | | | 0.093 | −0.068 | 904 | TreeC |
| D400 | 32.58 | 1.11 | 2.61 | | 0.051 | 0.113 | −0.088 | 882 | BldgL TreeC BldgH ImpS |
| D500 | 32.21 | 1.12 | 2.63 | | 0.050 | 0.153 | −0.099 | 894 | Water BldgH ImpS |

Specifically, the tree canopy width (TreeC) had a strong influence on temperature cooling at small scales. Within the range of D2.5–D10, a 10% increase in tree canopy width resulted in a temperature cooling of approximately 0.13°, 0.17°, and 0.24°. In slightly larger scales of D20 and D30 ranges, it led to a temperature reduction of about 0.36°. Within the larger scales of D40–D200, a consistent temperature reduction of approximately 0.5–0.6° can be achieved.

An increase in building area (BldgL) led to a rise in temperature, with the extent of its impact on temperature varying across different scale ranges. Within the range of

D150–D200, a 10% increase in building area resulted in a temperature rise of 0.14° to 0.23°, while within the range of D300–D400, the temperature increase ranged from 0.93° to 1.13°. This indicated that the cumulative effect of building area above D200 significantly influenced the temperature at the current point, causing it to increase. The temperature distribution in different zones in Figure 5a–d validated this finding as well: $Z_1$, with a dense tree canopy, had the lowest temperature at 32.2°, while $Z_4$, with the highest number of buildings, experienced the highest temperature at 39°. This further illustrated the impact of building area on temperature, which was consistent with the behavior of buildings above D150 under the sun condition.

Impervious surfaces (ImpS) exhibited a significant cooling effect on temperature within the large scale range. Within the range of D300–D500, a 10% increase in impervious surface area can lower the temperature by 0.68°, 0.88°, and 0.99°. Building height (BldgH) showed a negative correlation with temperature within the scale range of D20–D30, indicating that a 10-m building can lower the temperature by 0.14° and 0.12°. However, within the range of D400–D500, it showed a significant positive correlation, where an average building height of 10 m could increase the temperature by 0.5°.

Road distance (RoadD) did not have a significant impact on urban temperature under shadow conditions, and green space (GreenL), bare land (BareL), and water bodies (Water) showed no significant correlation with temperature at most scales. Hence, they were either not included in the model calculation or were not significant in regression models.

Exp. 3 (multi-scale) revealed (Table 6) the complex multiscale relationship between urban temperature variation and tree canopy, building height, impervious surface, and building area. At a small scale, increases in tree canopy and building height caused cooling, while at larger scales, increases in building area and height resulted in temperature rise. Moreover, at larger scales, the presence of impervious surface area showed a cooling effect. In comparison to Exp. 1, which examined single-scale impact factors, the multi-scale results demonstrated some discrepancies. For instance, the green space of D200 exhibited a cooling effect in the combined influence of D10 and large scale D200. Under multi-scale conditions, a 10% increase in tree canopy area in D10 and D20 stabilized the temperature decrease by 0.2 °C at the current point. The combined impact of D20 with larger scales indicated that a 10-m building height within D20 contributed to a temperature reduction of 0.25°, while an area percentage of 10% in building within D200 led to a temperature rise of 0.16 °C. Furthermore, a 10% increase in building area within D300 and D400 resulted in temperature increases of 0.77° and 1.01°. The multiscale analysis also demonstrated that the building density above D200 had a cumulative and additive effect on raising urban temperature.

**Table 6.** Multiscale analysis under shadow conditions: Stepwise linear regression results with significant parameter coefficients and AIC quality for each model.

| Scales | INT. | Time1 | Time2 | Significant Parameter | | | | AIC |
|---|---|---|---|---|---|---|---|---|
| D10 + D200 | 35.36 | 1.17 | 2.66 | TreeC-D10 −0.022 | GreenL-D200 −0.054 | | | 888 |
| D10 + D300 | 34.27 | 1.12 | 2.61 | TreeC-D10 −0.018 | BldgL-D300 0.078 | ImpS-D300 −0.060 | | 855 |
| D10 + D400 | 34.21 | 1.13 | 2.62 | TreeC-D10 −0.017 | BldgL-D400 0.101 | ImpS-D400 −0.071 | | 843 |
| D10 + D500 | 33.77 | 1.20 | 2.68 | TreeC-D10 −0.016 | TreeC-D500 −0.118 | BldgH-D500 0.076 | | 863 |
| D20 + D200 | 34.33 | 1.17 | 2.65 | TreeC-D20 −0.022 | BldgH-D20 −0.025 | BldgL-D200 0.016 | TreeC-D200 −0.041 | 875 |
| D20 + D300 | 34.01 | 1.13 | 2.62 | TreeC-D20 −0.021 | BldgH-D20 −0.018 | BldgL-D300 0.077 | ImpS-D300 −0.048 | 856 |
| D20 + D400 | 33.98 | 1.14 | 2.63 | TreeC-D20 −0.019 | BldgH-D20 −0.018 | BldgL-D400 0.101 | ImpS-D400 −0.061 | 845 |
| D20 + D500 | 33.28 | 1.19 | 2.67 | TreeC-D20 −0.020 | BldgH-D20 −0.015 | TreeC-D500 −0.102 | BldgH-D500 0.091 | 863 |

## 5. Discussion

We quantified the effects of urban spatial morphology indicators on atmospheric temperature at different scales, considering the influences of direct sunlight and shade, and paying attention to the interactions between scales. Critical values for the main influencing factors were identified based on the results of this study: in the city, the tree layout should focus more on the shading effect produced within 10 m, while the building layout should avoid excessively dense clusters of buildings beyond 150 m, especially within 300 m. Through a reasonable tree distribution and building layout, the local temperature increase in the city can be effectively alleviated.

### 5.1. Scale Dependence of Urban Temperature

This study has shown that urban temperatures exhibit multi-scale effects. Under direct sunlight, large-scale influences (150–300 m) dominate the temperature patterns. Conversely, under shadows, the combined effect of small-scale heterogeneity (10–20 m) and large-scale consistency (300–400 m) better determines temperature variations within the city. Specifically, spatial indicators within the 300-m scale determine the temperature's relative level at specific locations within the urban temperature distribution. Meanwhile, the 10-m scale governs temperature disparities among different locations at the same temperature level. These two scales provide crucial insights into understanding the similarity and variability in urban temperature distribution.

Therefore, previous studies focusing mainly on a single scale will not be able to adequately explain the formation mechanisms of temperature variations within cities. In contrast, the integrated approach of this study provides a more comprehensive understanding of urban temperature distributions by providing insight into the potential mechanisms of spatial correlation between such scales.

### 5.2. Influence of Urban Spatial Form on Atmospheric Temperature

We conducted detailed sampling and analysis of spatial form indicators at different scales within the city. Through precise ground observations and data processing, we were able to detect the impacts of different land surface types at various scales on the urban thermal environment. Tree canopies, impervious surfaces, and buildings are the primary influencing factors of urban atmospheric temperature variation, both under direct sunlight and shadow conditions. However, these factors may only show significant relationships at specific scales. For instance, tree canopies within 200 m have a notable effect on reducing local temperatures, while buildings exhibit a warming effect beyond 150 m. This indicates the importance of considering spatial form indicators across different scales and recognizing that their impact mechanisms may vary at different scales when studying urban temperature changes.

Furthermore, in this study, we drew different conclusions compared to previous research by distinguishing between trees and green space. It was found that trees can effectively reduce the temperature, with 100% tree coverage within the D10 scale contributing to a temperature reduction of approximately 2° (Table 6). Green space did not show a significant correlation with atmospheric temperatures, which differs from the findings of previous studies [36,46]. This difference can be attributed to the previous studies not differentiating between green space and trees, making it difficult to explore their temperature impacts at a finer scale. As a result, in urban design, careful consideration of how to plan and position urban trees to maximize the combined benefits of shading and transpiration becomes crucial.

In addition, under direct sunlight, we observed that small-scale impervious surfaces (D30) increased ambient temperatures, while under shadow conditions, large-scale impervious surfaces (D300–500) showed a cooling effect. This finding differs from many previous remote sensing studies that suggested that impervious surfaces would lead to higher surface temperatures [47–49].The reason behind this discrepancy lies in the fact that remote sensing studies often compare urban and rural temperature differences at a

macro scale. Rural areas mainly consist of bare land and vegetation, while urban areas are dominated by impervious surfaces, which have higher thermal emissivity and lower albedo, storing more heat and resulting in higher surface temperatures [50]. Additionally, the low spatial resolution of RS temperature inversion, combined with the prevalence of sunlit areas in urban regions compared to shaded areas, means that more information about the temperature under direct sunlight is reflected within a pixel. This leads to a tendency to observe a positive correlation between impervious surfaces and surface temperature.

However, our research observed a cooling effect of large-scale impervious surfaces under shaded conditions. This can be attributed to the abundance of trees in the four sampled areas of our study (excluding commercial center $Z_4$), especially in the university campus $Z_1$, where tree cover exceeds 50% and impervious surfaces account for 50%. These trees are predominantly planted on impervious surfaces. Under shadow conditions, the trees provide shade and transpiration, reducing ambient temperatures and increasing the cooling effect of impervious surfaces on a large scale. This explanation aligns with the findings of [10], whose research indicated that in their study area, for 25% of impervious surface locations (e.g., most residential areas), temperature decreased most rapidly when canopy coverage exceeded 40%. Therefore, our study emphasizes the need for further investigation into the interaction between impervious surfaces and trees to understand the complex mechanisms of urban temperature variations.

Additionally, the height and density of buildings have been a focal point of numerous studies. This study reveals that the main reason for urban temperature rise is the aggregation effect caused by buildings with a size of more than 150 m (D150 and above). This phenomenon can also explain the urban heat island effect, which is generated by the cumulative increase of buildings taller than 150 m. Of particular concern is the significant warming effect caused by the cumulative impact of buildings with a size of D300 and above (for example, within D10 + D300 m/D400 m, 10% increase of buildings will raise the temperature by 0.78°/1°, as shown in Table 6). It is worth noting that this research specifically quantified the relationship between building height and temperature within different scales. At the small scale (D10–20 m), the study found that building height has a cooling effect on summer temperatures. This is because tall buildings can enhance the efficiency of land–air convection, accelerating air circulation and increasing shading on the ground, leading to reduced solar radiation absorption. Thus, under shadow conditions, the combined effect of building shadows and canopy coverage results in a cooling phenomenon. However, at the large scale (D400–500 m), the increase in building area leads to the heat island effect. Specifically, as the building area around a point increases, the warming effect becomes more pronounced, counteracting the cooling effect caused by local tall building shadows. This phenomenon can be attributed to the different effects of various factors and mechanisms at different scales [51], which explains seemingly contradictory results in the studies of [16,52]. While Huang's research at the small scale indicates a cooling effect of building height on summer temperatures, Zhou's study at the large scale suggests a warming effect of building height on summer temperature.

Therefore, this study comprehensively considers the influence of different factors under sunlit and shadow conditions, delving into the relationship between various spatial form indicators and atmospheric temperature at multiple scales. Understanding the threshold and quantified contributions of each influencing factor to temperature is crucial for improving urban climate.

### 5.3. How Should Urban Landscape Be Laid Out to Most Effectively Cool the City?

The urban thermal environment (UTE) effect is influenced by multiple factors, and its formation mechanisms vary from micro to macro scales. Therefore, it is essential to consider a comprehensive set of factors to devise effective mitigation strategies. In meticulous urban planning, special attention should be given to buildings, trees, and impervious surfaces. We suggest urban planners control building height and density as crucial measures to mitigate the UTE. Reducing the concentration of buildings over 150 m, particularly those beyond

300 m, can lower the formation of heat islands. Additionally, in low-density building areas, vertical expansion of buildings instead of horizontal expansion can reduce daytime urban heat environments.

Increasing tree canopy cover in urban areas is an effective measure to improve localized microclimates. In urban planning, tree layout and planting should be carefully considered, especially in densely built areas. Planting large-canopy trees or increasing planting density within every 10 m can alleviate the warming effect caused by buildings. Moreover, prioritizing tree planting in sun-exposed areas, rather than dispersing trees around buildings, can better utilize the shading and transpiration effects of trees.

Impervious surfaces in the city, such as concrete and asphalt, absorb and release heat under direct sunlight, exacerbating the heat island effect. To lower temperatures, it is advisable to disperse impervious surfaces and appropriately distribute trees around them. This can effectively reduce the temperature of the surrounding areas through the transpiration and shading effects of trees.

*5.4. Limitations*

In our experimental design, we carefully considered factors influencing the selection of the measurement time period. Due to relatively minor temperature variations in the morning, we opted to conduct experiments during the mid-day and later hours. While our research findings have provided valuable insights into the thermal environment mid-day, it is important to emphasize that these results pertain specifically to this particular time window. Future studies should consider expanding the time range to include data from the afternoon and evening, enabling a more comprehensive understanding of urban thermal dynamics.

## 6. Conclusions

This paper quantitatively investigates the impacts of factors influencing the UTE at different scales and determines the comprehensive effects between spatial scales to better understand the characteristics and formation mechanisms of urban heat islands. The study focuses on the effects of urban shading and spatial form features on the thermal environment at various scales. The following conclusions are drawn:

(1)   Research on urban heat environment requires comprehensive consideration of influencing factors at different scales to better understand the mechanisms responsible for urban heat environments formation.

(2)   Tree canopy cover, impervious surfaces, and buildings are the primary factors influencing the urban heat environment, and their critical thresholds of impact have been clearly identified.

(3)   Trees play a significant role in temperature reduction, particularly under direct sunlight and shadow conditions, with the cooling effect dependent on the canopy coverage ratio. Notably, no significant correlation was observed between temperature and green space.

(4)   Impervious surfaces exhibit a significant cooling effect under shadow conditions. The impact of impervious surfaces on temperature is altered by the presence of shadows, and the final effect depends on the proportion of impervious surfaces covered by shadows.

(5)   Building density and height demonstrate a significant correlation with temperature. Increasing building density leads to higher temperatures during summer days, while within smaller scales, taller buildings contribute to temperature reduction.

In summary, this study conducted a comprehensive quantitative analysis, taking into account the influence of urban shadows and spatial form features on the mid-day urban thermal environment. However, due to limitations in data collection time, our research primarily focused on the mid-day period (12:00 to 13:30) during three summer days. Therefore, our research findings are primarily applicable to this specific time window. Other time periods may introduce additional complexities, such as variations in atmospheric temperature exchange due to different shadow positions, which require further investigation.

We recognize that UHI effects can vary throughout the day, and the role of tropical nights in the urban thermal environment is also significant. Future research should consider expanding the data collection to include additional time periods for a more comprehensive understanding of UTE dynamics.

Nevertheless, our study provides valuable insights into the quantitative relationship between urban spatial morphology and midday UHI intensity. It offers scientific guidance to urban planners for managing and mitigating UTE at various scales during the mid-day hours. For a more comprehensive understanding of UTE throughout the day and night, further research is needed to explore these additional time frames and their unique characteristics.

**Author Contributions:** Conceptualization, L.L.; methodology, L.L.; investigation, L.L.; writing—original draft preparation, Y.D.; writing—review and editing, L.L.; visualization, Y.D.; supervision, M.P.; funding acquisition, L.L.; data curation, M.P.; validation, L.Z. and S.Q. All authors have read and agreed to the published version of the manuscript.

**Funding:** This research was supported by the National Natural Science Foundation of China (Grant No.32001368).

**Data Availability Statement:** Not applicable.

**Acknowledgments:** The authors would like to thank the editors and reviewers for providing their valuable comments and suggestions.

**Conflicts of Interest:** The authors declare no conflict of interest.

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
