# Peer review of "Multi-Scale Influence Analysis of Urban Shadow and Spatial Form Features on Urban Thermal Environment"

_remotesensing, doi:10.3390/rs15204902_

Round 1

Reviewer 1 Report

The study, titled " Multi-scale influence analysis of Urban Shadow and Spatial Form Features on Urban Thermal Environment," investigates the contributions made by building and tree shadows to the urban thermal environment. The authors collected data using various thermal and UAV drone equipment. The results were analyzed using various statistical techniques, such as the ANOVA test. The primary focus of the study was on four different land cover types: university areas, parks, residential areas, and commercial areas. One interesting conclusion of the study is the correlation between the temperature and building area and height. Building area exhibits an inverse correlation while building height shows a direct correlation.

The paper is well-written and logically presented. The methods and experimental details are explained thoroughly. In addition to clear and detailed representations, the results are well-discussed. While the paper may not make a significant scientific contribution, it holds value for urban planners and researchers studying urban heat islands. Therefore, I recommend accepting this study in its present form.

Author Response

Thanks very much for your insightful comments. We have made minor revisions according to other reviewers' suggestions. The revisions are highlighted in the manuscript.

Reviewer 2 Report

This paper addresses how urban thermal environment (UTE) effects manifest at specific spatial scales whose radii ranging from 2.5 to 500 m. For each measuring location, the defined scales were featured by different urban indicator parameters such as land cover in the form of buildings, green spaces, impervious surfaces, asphalt, water bodies and bare ground, and other miscellaneous parameters such as tree canopy percentage and average building height. The study also focuses on how these effects manifest at urban shading sites in addition to the direct solar irradiated locations. The multi-scale effects of urban morphological indicators on air temperature are investigated with the use of a linear mixed-effects model methodology in combination with the Akaike information criterion (AIC) to assess model accuracy for optimal selection of best model performance.

The paper stands for a good quality research, and it is presented in an almost final form, probably due to previous revisions before the present one. The topic covered reveals a modern-day interest for urban heat inland research.

Even that the paper could be accepted in its present form, some questions arise with their respective recommendations that authors should be clarified or correct to the editor’s satisfaction:

Figure 1a – I consider that this partial figure may contain political issues. I strongly recommend avoiding it. Instead, the authors may use a map containing the several countries in the region indicating Wuhan location.

Lines 160-161 – The instrumental campaign run for three summer days from 12:00 to 13:30. This should be considered as a limitation of the study and must be reflected in the title, abstract, discussion and conclusions. It is obvious that paper’s findings rule only for the midday period where shadow arrangement do not necessarily coincide with afternoon or morning outline. On the other side, it is true that UHI effects are more important at midday, but also the role of tropical nights is also crucial for the urban thermal environment. All these questions should be addressed in the paper at their respective sections.

Line 165 – The authors state that measurements were acquired at the “height of 0.8 meters above the ground to ensure minimal interference and a stable measurement environment”. This is a quite short height since normal air temperature samples should be made at 1.5m to 2.0m regarding WMO. Setting measurements at 0.8m high do not invalidate the study, but the authors should justify more exhaustively why they decided to employ this height in opposition to WMO recommendations.

Lines 169-178 – Please, include the explanations given here in a new table providing the information of the three sampling sessions.

Line 237 – AIC is a relative quality predictor for a given set of candidate models, being the preferred model the one with the minimum AIC. So, I would not use the term “accuracy”. Use instead “quality” for example, since it is a relative magnitude that depends on the set of models chosen.

Figure 5 a and b – Use Exp.1 and Exp.2 instead of Lab1 and Lab2 at the legend. Entitle y-axes as "AIC quality" and set units if appropriate.

Figure 5 c – A title for the Colorbar palette should be used, i.e. AIC quality.

Figure 5 d – Better than representing prediction values over the original, provide model's residuals to check they are distributed randomly as a test of the model's predictions.

Tables 3, 4 and 5 – Captions should contain what the table entries are, more than simply specifying them as "regression results".

Please check minor errors and spaces after punctuation marks.

Reviewer 3 Report

Introduction and references

One of the most referenced subjects in the current scientific literature is Urban Heat Island.

The authors use recent references, which could be extended to the analysis of this topic over a longer period of time. One reference that I recommend is that of M. Santamouri, entitled Heat island research in Europe The state of the art. Advances in Building Energy Research, 1 (2007), pp. 123-15. This author has other very interesting references. Also, some works in China such as those by Yang, et al, entitled, PM2.5 Pollution Modulates Wintertime Urban Heat Island Intensity in the Beijing-Tianjin-Hebei Megalopolis, China, or The influence of 2D/3D urban spatial form indicators on surface Urban Heat Island based on Spatial Regression Models: A case study of Hangzhou, China, Chen, Haotian ; Zheng, Sheng

When it says "several studies have examined the Impact..." on line 100, it should have more references, not just two.

In synthesis: the introduction should be revised in order to have more references, especially with a longer time frame.

Graphic semiology.

In figure 1. The authors have integrated maps and graphics into a single composition. While the idea may be good, the result is confusing and in some cases unreadable. I recommend that the two graphics (a) and (b) be separated from the composition and presented individually with more quality. I recommend that the main map has a specific legend in UTM, the grid coordinate reference system, at this scale, is not very useful and legible.

Methodology

I consider that a table (graph) with the working methodology is necessary.

Formal aspects

The tables need to be revised. There are formal aspects to correct. For example: Table 4.Single.... it is necessary to separate the text from the full stop; the names of the items in two lines..., this causes a complex reading of the tables. Reading is also necessary to correct separations of commas, full stops, etc.
